# Objective and Subjective Assessment of Music Perception and Musical Experiences in Young Cochlear Implant Users

Miryam Calvino [1,2], Alejandro Zuazua-González [3], Javier Gavilán [1] and Luis Lassaletta [1,2,*]

[1] Department of Otorhinolaryngology, Hospital La Paz, IdiPAZ Research Institute, 28046 Madrid, Spain; miryam.calvino@salud.madrid.org (M.C.); javier.gavilan@salud.madrid.org (J.G.)

[2] Biomedical Research Networking Centre on Rare Diseases (CIBERER), Institute of Health Carlos III (CIBERER-U761), 28029 Madrid, Spain

[3] Department of Otorhinolaryngology, Hospital Infanta Leonor, 28031 Madrid, Spain; alejandro.zuazua@salud.madrid.org

[*] Correspondence: llassaletta@salud.madrid.org

**Abstract:** For many individuals, music has a significant impact on the quality and enjoyability of life. Cochlear implant (CI) users must cope with the constraints that the CI imposes on music perception. Here, we assessed the musical experiences of young CI users and age-matched controls with normal hearing (NH). CI users and NH peers were divided into subgroups according to age: children and adolescents. Participants were tested on their ability to recognize vocal and instrumental music and instruments. A music questionnaire for pediatric populations (MuQPP) was also used. CI users and NH peers identified a similar percentage of vocal music. CI users were significantly worse at recognizing instruments ($p < 0.05$) and instrumental music ($p < 0.05$). CI users scored similarly to NH peers on the MuQPP, except for the musical frequency domain, where CI users in the children subgroup scored higher than their NH peers ($p = 0.009$). For CI users in the children subgroup, the identification of instrumental music was positively correlated with music importance ($p = 0.029$). Young CI users have significant deficits in some aspects of music perception (instrumental music and instrument identification) but have similar scores to NH peers in terms of interest in music, frequency of music exposure, and importance of music.

**Keywords:** cochlear implant; music; songs; instruments; questionnaire; children; adolescents





## 1. Introduction

The central objective of early cochlear implantation in childhood is to facilitate the development of spoken language [1]. Despite this primacy of language development, the development of other auditory skills should not be neglected. One such auditory skill is music perception and appreciation.

Music is believed to be a ubiquitous cultural phenomenon and can be experienced both passively and through active participation [2]. Children and adolescents participate in music by, e.g., dancing, singing, or playing an instrument [3]. Music is related to a greater quality of life and satisfaction with health [4], and an increasing body of evidence suggests beneficial results of music interventions for children with hearing loss [5].

Even for individuals with good auditory rehabilitation, music can nevertheless be challenging as it is perhaps the most complex auditory stimulus that exists [6]. Music thus provides an excellent probe to evaluate the technological, biological, and acoustic limitations related to the use of cochlear implants (CIs) [7], as well as providing a challenging stimulus for rehabilitation. By focusing training on particular musical attributes such as pitch and timbre perception [8], CI users may be able to improve their perception of sounds in general.

Moreover, beyond pitch and timbre perception are several phenomena that are also important in the experience of listening to music, such as subjective quality, mood, and

the situational context [9]. However, it is easy to think that these aspects of the musical experience are common for CI users and normal hearing (NH) individuals.

It is necessary to identify precise methods for testing the individual variation in CI users' music perception. In the literature, several tools have been used to check differences in music perception among CI users [10]. Although there is no agreement about the best way to measure it, the importance of a multidisciplinary research [11,12] in this area including both subjective and objective methods is well known.

Here, we used both objective and subjective assessments to investigate how pediatric CI users with prelingual deafness interact with music. We thought this evaluation could help to predict speech performance in this group of participants. With objective testing, we evaluated the abilities of users to identify vocal and instrumental music and instruments. For subjective assessment, we used a self-reported music questionnaire to probe musical participation, musical interests, frequency of music exposure, and the importance of music. These assessments were compared with those of an age-matched cohort of NH peers. The correlations between the objective and subjective results were also analyzed.

## 2. Materials and Methods

### 2.1. Participants

This cross-sectional study took place between November 2020 and October 2022. It was approved by the ethics committee of the University Hospital of La Paz (approval number: HULP PI-4447) and was registered with ClinicalTrials.gov (trial identifier: NCT05319678).

The criteria for inclusion in the CI group were individuals with prelingual deafness who underwent cochlear implantation before the age of 3 years [13] and who were aged between 6 and 16 years [14] at the time of the evaluation. All participants were required to be unilaterally or bilaterally implanted with a MED-EL CI (MED-EL GmbH, Innsbruck, Austria), with at least 10 active electrodes, at least one year of stable fitting, and with good speech performance (defined as ≥65% of correctly identified disyllables in silence).

The criteria for inclusion in the NH control group were children with normal hearing (NH) (PTA ≤ 30 dB at 0.5, 1, and 2 kHz) who were aged between 6 and 16 years at the time of the evaluation. They were recruited from hospital staff's relatives after performing an audiometric test that confirmed their healthy hearing status. For both groups, fluency in the Spanish language was required. Criteria for exclusion from participation were the presence of additional any medical disabilities (beyond hearing loss in the CI users).

For both groups, the enrolled participants were divided into two subgroups based on their age at the time of evaluation: children (6–10 years) and adolescents (11–16 years).

### 2.2. Vocal vs. Instrumental Music Recognition and Instrument Identification

The ability of participants to discriminate popular songs (vocal music) and their melodies (without the lyrics, e.g., instrumental music), as well as to discriminate different musical instruments, was studied (Figure 1).

Based on previous interviews with subjects from all age groups, we created two different music playlists for the children and adolescent subgroups. They were created based on the Spotify playlist for CI users designed by MEDEL (https://open.spotify.com/user/medelcochlearimplants, accessed on 11 January 2024). For the children subgroup, the playlist consisted of 15 of the most popular songs for children. For adolescents, the list of 30 songs chosen were based on their preferences according to their age (see Supplementary Material).

Prior to the start of the study, each participant with the "pencil and paper" method had to show us which songs they were familiar with. During testing, the participants listened to 10 randomly selected songs. They then had to identify which song it was and mark it on the list. If the song was not identified within 60 s, it was considered as an incorrect response. The percentage of correctly recognized songs was recorded for each participant.

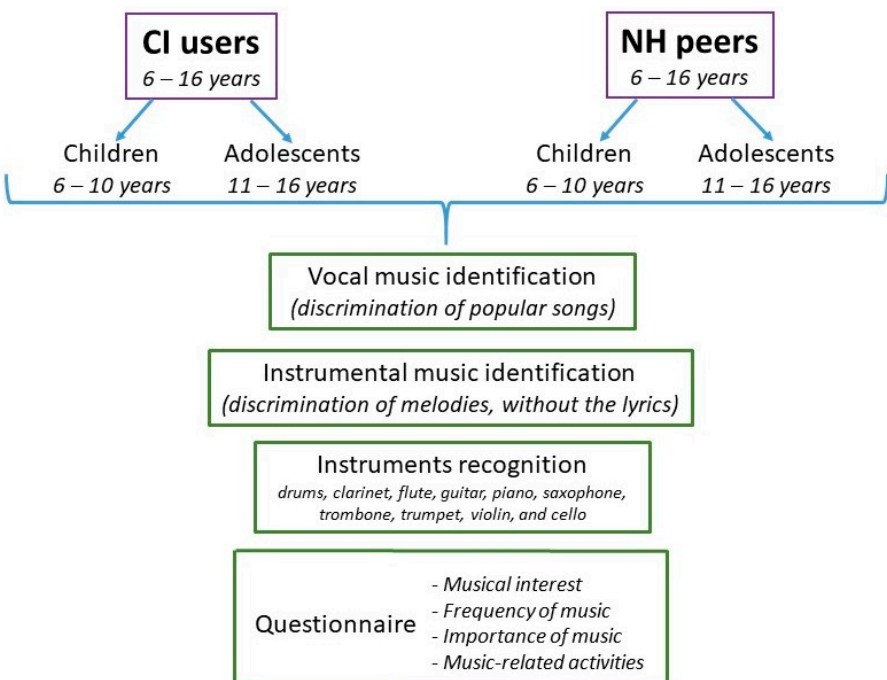

**Figure 1.** Flow chart of the experimental protocol. Participants and tests used are displayed.

For CI users, the direct audio input (DAI) cable was used to connect the audio processor to the sound source of the computer in order to avoid any hearing with the contralateral ear in individuals with residual hearing. For the NH participants, headphones were used.

The same procedure was used to evaluate instrumental music recognition; that is, participants were evaluated on the songs without the lyrics. The lyrics were digitally extracted.

For instrument identification, both groups were asked to identify 10 commonly known instruments: drums, clarinet, flute, guitar, piano, saxophone, trombone, trumpet, violin, and cello. A range of low, middle, and high frequencies are represented by these instruments. As conducted by others [15], professional musicians recorded chords for the sound clips. The evaluation was performed in the same manner as with the vocal and instrumental music identification task. If the instrument was not identified within 60 s, it was considered an incorrect response. The percentage of correctly recognized instruments was recorded for each participant.

### 2.3. Subjective Assessment with the MuQPP

After reviewing the existing literature, we did not find any questionnaire suitable both in language and in the form for subjects aged 6–16 years. Therefore, we created a music questionnaire for the pediatric population (MuQPP) (see Supplementary Material). We wished to create a tool similar to the music-related quality of life questionnaire (MuRQoL) [16], which has been already validated in the Spanish language for us with adults (article in press).

The MuQPP is comprised of a set of 13 main questions which have already been used by other authors with children and adolescents [17–19]. These questions are divided into four domains (Figure 1):

(1) Musical interests (e.g., "Do you play a musical instrument?"). Answer options: yes (3 points), sometimes (2 points), no longer (1 point), and no (0 points).
(2) Frequency of music (e.g., "How often do you sing?"): scored on a 7-point Likert-type scale.
(3) Importance of music (e.g., "How important is music in your life?"): scored on a 7-point Likert-type scale.
(4) Music-related activities (e.g., "Do you perform any musical activity?"). Answer options: yes/no.

*2.4. Statistical Analysis*

Demographic characteristics and outcome measures are shown as absolute numbers (*n*) and relative frequencies (%), and if appropriate, as mean with standard deviation ($\pm$SD) and range.

For the scores on the objective and subjective assessments, normality testing was performed with the Kolmogorov–Smirnov test. Data were found to be not normally distributed even after value transposition to log(x) and exp(x). Therefore, both intergroup (CI vs. NH) and intragroup (pediatric population both in CI users and NH) comparisons were evaluated with the Mann–Whitney U test. Fisher's exact test was used to compare the number of participants who correctly identified each vocal and instrumental music and instruments, as well as differences in the questionnaire between both groups. Spearman correlation coefficients were calculated to assess the relationships between mean scores on the objective and subjective assessment.

Missing data were treated as missing values. A level of $p \leq 0.05$ (2-tailed) was considered statistically significant. Statistical analyses were performed in SPSS version 24.0 (IBM, Armonik, NY, USA).

**3. Results**

*3.1. Subjects*

Thirty CI users participated in this study. A total of 14 were assigned to the children subgroup and 16 to the adolescent subgroup. The same number of age-matched NH controls participated in each subgroup. The demographic information of both groups is given in Table 1.

**Table 1.** Demographic information of all participants and implantation characteristics of the CI users. Participant age and CI duration are given as mean years $\pm$ standard deviation, with ranges in parentheses. All other values are given as absolute number with percentage in parentheses. N/A, not applicable. [1] In cases of sequential bilateral implantation, the time of the first CI is given. [2] In cases of bilateral CI users, the newest audio processor is given.

| | CI Users | | NH Peers | |
|---|---|---|---|---|
| | **Children** | **Adolescents** | **Children** | **Adolescents** |
| Total participants | 14 | 16 | 14 | 16 |
| Male | 4 (29%) | 9 (56%) | 6 (43%) | 8 (50%) |
| Female | 10 (71%) | 7 (44%) | 8 (57%) | 8 (50%) |
| Age at evaluation | 8 $\pm$ 1 (6–10) | 14 $\pm$ 1 (12–16) | 8 $\pm$ 1 (6–10) | 13 $\pm$ 1 (11–16) |
| CI duration of use [1] | 7 $\pm$ 1 (5–9) | 12 $\pm$ 1 (10–14) | N/A | N/A |
| Type of implantation | | | | |
| Unilateral | 0 (0%) | 1 (6%) | N/A | N/A |
| Bilateral | 13 (93%) | 15 (94%) | N/A | N/A |
| Bimodal | 1 (7%) | 0 (0%) | N/A | N/A |
| Audio processor [2] | | | | |
| OPUS 2 | 1 (7%) | 3 (19%) | N/A | N/A |
| RONDO | 4 (29%) | 0 (0%) | N/A | N/A |
| SONNET | 2 (14%) | 10 (63%) | N/A | N/A |
| RONDO 2 | 6 (43%) | 2 (12%) | N/A | N/A |
| SONNET 2 | 0 (0%) | 1 (6%) | N/A | N/A |
| RONDO 3 | 1 (7%) | 0 (0%) | N/A | N/A |

*3.2. Vocal and Instrumental Music Recognition and Instrument Identification*

Table 2 shows the percentage of vocal music, instrumental music, and instruments correctly identified by participants.

For CI users, vocal music was correctly identified by 98 $\pm$ 6% of children and 93 $\pm$ 17% of adolescents. All NH peers correctly identified all songs. There were no significant

differences between CI users and NH peers in vocal music identification ($p = 0.709$ and 0.316 for children and adolescents, respectively).

**Table 2.** Mean and standard deviation of the percentage of vocal and instrumental music and instruments correctly identified by all groups of participants. Comparisons which were significant according to Fisher's exact test are given in italics.

|  | Children | | | Adolescents | | |
|---|---|---|---|---|---|---|
|  | **CI Users** | **NH Peers** | ***p*** | **CI Users** | **NH Peers** | ***p*** |
| **Vocal music (%)** | $98 \pm 6$ | 100 | 0.709 | $93 \pm 17$ | 100 | 0.316 |
| **Instrumental music (%)** | $60 \pm 19$ | $93 \pm 11$ | <0.0001 | $47 \pm 27$ | $97 \pm 6$ | <0.0001 |
| **Instruments (%)** | $55 \pm 25$ | $79 \pm 22$ | 0.036 | $58 \pm 19$ | $85 \pm 14$ | <0.0001 |

For CI users, instrumental music was correctly identified by $60 \pm 19\%$ of children and $47 \pm 27\%$ of adolescents. For NH peers, instrumental music was correctly identified by $93 \pm 11\%$ of children and $97 \pm 6\%$ of adolescents. CI users were significantly worse than NH peers and differed significantly at instrumental music identification ($p < 0.0001$ for both).

For CI users, instruments were correctly identified by $55 \pm 25\%$ of children and $58 \pm 19\%$ of adolescents. For NH peers, instruments were correctly identified by $79 \pm 22\%$ of children and $85 \pm 14\%$ of adolescents. CI users were significantly worse than NH peers at instrument identification for both the children ($p = 0.036$) and adolescent subgroups ($p < 0.0001$).

There were differences in the adolescent subgroup between CI users and their NH peers regarding the percentage who identified each instrument correctly (Figure 2). For the adolescent subgroup, CI users were significantly worse at identifying the clarinet, the flute, and the violin ($p = 0.001$, 0.01, and 0.022, respectively). No other differences were significant in the adolescent subgroup, nor were any differences significant in the children subgroup.

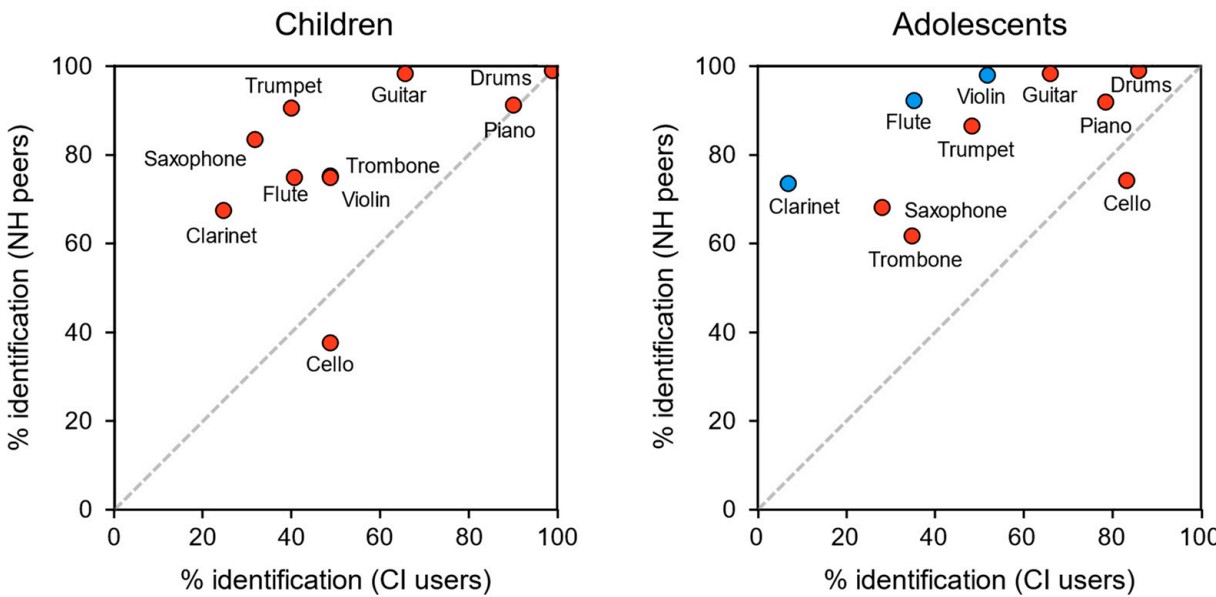

**Figure 2.** Comparisons of instrument identification among CI users and NH peers. Mean percentage identification rates are given. Values above the diagonal indicate superior identification by NH peers. Instruments with statistically significant differences in identification rates are given in blue.

When considering the two groups without regard to age (i.e., all CI users vs. all NH peers), CI users were significantly worse at identifying the saxophone, trombone, trumpet, and violin ($p = 0.039$, 0.031, 0.046, and 0.02, respectively).

For CI users as a whole, the drums (93% correct identification) followed by the piano (85% correct identification) were the easiest instruments to identify. The saxophone (31%) and clarinet (15%) and were the most difficult to recognize.

Among both children and adolescents, CI users were slightly better than NH peers at identifying the cello, although this difference was not significant in either age subgroup.

### 3.3. Musical Questionnaire Responses

Both children and adolescents with CIs scored similarly to their NH peers on the musical interests, frequency of music, and the importance of music domains. The exception was CI users in the children subgroup, whose mean score in the frequency items was higher than their NH peers ($54.2 \pm 12.9$ vs. $40.9 \pm 12.1$, $p = 0.009$). Forty percent of NH peers in the adolescent subgroup reported that they participate in music-related activities, while none of the CI users in the adolescent subgroup did ($p = 0.017$) (Table 3).

**Table 3.** Self-reported participation in music-related activities (e.g., playing the guitar or piano, musical theatre, playing in a musical group, etc.) among CI users and NH peers, and value of Fisher's exact test.

|  | Children | | Adolescents | |
|---|---|---|---|---|
|  | No | Yes | No | Yes |
| **CI users** | 64% | 36% | 100% | 0% |
| **NH peers** | 50% | 50% | 60% | 40% |
|  | $p = 0.063$ | | $p = 0.017$ | |

Within the CI group, children scored higher than adolescents on the musical interests domain ($16.1 \pm 3.8$ vs. $12.2 \pm 5.1$, $p = 0.035$). Thirty-six percent of CI users in the children subgroup participated in music-related activities, while none of adolescent CI users did ($p = 0.027$) (Table 3).

### 3.4. Correlations between Objective and Self-Reported Outcomes

In the children subgroup, a significant correlation was observed between the melody identification task and the importance of music domain of the MuQPP among CI users ($p = 0.028$) (Table 4). A significant correlation was also observed between the melody identification task and the musical interests domain of the MuQPP among NH peers ($p = 0.009$).

**Table 4.** Pairwise Spearman correlation coefficients between scores on the three domains of the music questionnaire for pediatric population (MuQPP) and the percentages of correctly identified items in the vocal and instrumental music identification and instrument identification tasks. Spearman coefficients are given above; *p*-values are given below. Significant correlations are given in italics. No pairwise correlations are given with the vocal music identification task for the NH peers as all NH peers scored 100% in this task. Data are shown for the children subgroup only.

|  | CI Users | | | NH Peers | | |
|---|---|---|---|---|---|---|
|  | Musical Interests | Frequency of Music | Importance of Music | Musical Interests | Frequency of Music | Importance of Music |
| **Vocal music (%)** | −0.431 | 0.064 | −0.231 | - | - | - |
|  | 0.213 | 0.861 | 0.521 | - | - | - |
| **Instrumental music (%)** | 0.612 | 0.532 | 0.688 | 0.668 | 0.180 | 0.296 |
|  | 0.060 | 0.114 | *0.028* | *0.009* | 0.537 | 0.304 |
| **Instrument (%)** | 0.269 | 0.454 | 0.252 | −0.217 | −0.164 | 0.213 |
|  | 0.399 | 0.138 | 0.430 | 0.456 | 0.574 | 0.465 |

For the adolescent subgroup, no significant correlations were observed between any tasks and scores for any domain of the MuQPP.

## 4. Discussion

In this study, we evaluated music perception skills and musical interests in a cohort of young CI users and age-matched peers. It was observed that overall, CI users were as adept as their NH peers at identifying popular songs. On the instrumental music recognition and instrument identification tasks, CI users faired significantly worse overall. Although CI users had worse performance on the identification of most instruments, these differences were significant only for the clarinet, flute, and trombone. On the self-reported MuQPP, scores for the domains of musical interests, frequency of music exposure, and the importance of music were largely similar between CI users and their NH peers. This study shows how young CI users with prelingual deafness subjectively enjoy music to a similar degree as their NH peers but have significant deficits in some aspects of musical perception, namely, instrumental music recognition and instrument identification.

### 4.1. Vocal and Instrumental Music Recognition

CI users had song identification (vocal music) rates of over 90%. High levels of musical skill among young CI users were also observed in a recent publication, where users showed similar abilities to their NH peers in most of the evaluation categories of the online musical training platform Meludia [20].

Music perception can be assessed by means of several different tests, such as the intelligibility of the lyrics. Some studies have observed that CI users make use of the lyrics in music to identify and enjoy music [21], as well as to compensate for poor timbre, pitch, and melody identification [3].

As described in the results section, children and adolescent CI users were able to recognize only around half of the presented instrumental music compared to NH participants who identified a percentage close to 95%. This finding is in accord with other publications. Hsiao et al. showed that children aged 7–15 years who speak Mandarin Chinese (a tonal language) had better results on familiar melody recognition when lyrics were present [22]. CI users with prelingual deafness have average discrimination thresholds of 2–3 semitones, in contrast to 0.1–0.2 semitones for the NH peers [23]. This likely underlies the poorer performance of CI users in melodic contour identification [24]. Buyens et al. concluded that enhancing vocals by 6 dB relative to the background instruments led to greater enjoyment of music by CI users [25]. Accordingly, it was hypothesized that reducing the complexity of the backing music may assist the noise reduction algorithm of the CI sound coding strategy [21].

### 4.2. Instrument Identification among CI Users

The abilities of CI users to identify commonly known instruments has been previously studied [26]. This skill has been ranked as the third most valuable for music perception [27]. CI users in the present study were more successful at identifying the drums and the piano than other instruments. Similarly, Looi and She observed that around 70% of adult CI users could always identify the piano and drums [27].

In the study of Looi and She, CI users seemed to express a preference for low-frequency instruments. In the present work, there was a considerable difference in the recognition rates of the violin and the cello among adolescent CI users. Just over half recognized the violin, while over 80% recognized the cello. This is despite the physical similarity of these instruments, both being members of the violin family. Perhaps the most obvious difference between these two instruments is the note range, with the cello being an octave and a fifth lower than the violin.

Kim et al. observed that young adolescent CI users had higher recognition rates for the piano and the guitar [28]. They proposed that the distinct qualities of each instrument (the rapid attack time of the piano and the distinctive temporal envelopes of the guitar)

were key to discriminating these instruments from others. After the drums and piano, the guitar was the next most recognized instrument by the CI users in the present study. When comparing CI users with NH peers, Looi et al. found that higher frequency instruments (e.g., flute and violin) had a noisier and duller quality of sound [27]. This finding could be in agreement with our results, which revealed that NH participants identified more musical instruments than CI users.

Interestingly, we found that adolescents with CI were worse than their NH peers at identifying the clarinet, flute, and violin. This finding could seem odd since these are quite different instruments, and a priori, the differences should have observed between bass and non-bass instruments, as seen in some of the referenced previous studies. A possible explanation for this could be that individual CI subjects may exhibit different patterns of results across instruments, possibly influenced by their relative "pleasantness" [29], i.e., they cannot identity an instrument if they have not listened to it previously. As some authors have already mentioned [3,30], music training may help to improve music perception.

### 4.3. The Value of Subjective Music Assessment

Objective tests are useful for assessing musical perceptual abilities but give no insight into the actual exposure or importance of music in everyday life. Subjective questionnaires such as the one used here can provide such insights [21,31].

Music is a universal language, a part of young people's everyday life. It is claimed that young people perceive music as "providing a sense of companionship" [32]. As such, it is perhaps unsurprising that young CI users and their NH peers had similar scores for musical interest and the importance given to music. Previous studies have shown that music preferences among young people are influenced by their relationships with peers and social identities within peer groups [33]. Young CI users and their NH peers share musical environments during their music preference development phase [28], a fact which is in accord with the similar scores on the questionnaire used in our study.

### 4.4. Limitations and Possible Applications of this Study

Our study was not limitation-free. First, the single-center design of this study could limit the external validity of the outcomes. However, as the inclusion criteria were not restrictive, we believe that the results may apply to any population.

Moreover, it is important to remark that the scale used to evaluate music habits was not validated or adapted for our culture and language. However, despite this limitation, correlations studied were similar to those in previous studies. Other manuscripts have also used no validated questionnaires [3].

Music rehabilitation, education, and training offer the opportunity for young CI users to develop their musical skills [30]. Due to the cross-sectional design of our study, we did not evaluate longitudinal outcomes after a period of musical training. It would be of benefit to track the measures reported here longitudinally to observe their dynamics with prolonged CI use. Yüksel et al. [3] affirmed that the relation between music recognition skills, music enjoyment, and listening habits in young CI users with prelingual deafness can be bidirectional: the higher music exposure, the greater the musical recognition skills. Prolonged contact with music may also help with neural adaptation to music processing [34].

In line with this music training, this study might encourage CI young users (and even extended to other age groups) to listen to music in order to enhance the benefit obtained with their CI(s).

### 5. Conclusions

The findings of this study suggest that young CI users with prelingual deafness have perceptual deficits for some musical attributes (instrumental music recognition and instrument identification). Nevertheless, they give similar subjective ratings of musical

interests, frequency of music exposure, and the importance of music in their lives to those of their peers with normal hearing.

**Supplementary Materials:** The following supporting information can be downloaded at: https://www.mdpi.com/article/10.3390/audiolres14010008/s1.

**Author Contributions:** Conceptualization, M.C., J.G. and L.L.; methodology, M.C.; software, M.C. and A.Z.-G.; formal analysis, M.C.; investigation, M.C. and A.Z.-G.; data curation, M.C. and L.L.; writing—original draft preparation, M.C.; writing—review and editing, M.C. and L.L.; supervision, L.L.; funding acquisition, L.L. All authors have read and agreed to the published version of the manuscript.

**Funding:** This work was supported by a grant (Grant Number: PI21/00147) from the Programa Estatal de Generación de Conocimiento y Fortalecimiento del Sistema Español de I + D + I, ISCIII, Spain.

**Institutional Review Board Statement:** This study was conducted in accordance with the Declaration of Helsinki and approved by the Ethics Committee of La Paz University Hospital (protocol code HULP PI-4447, 5 November 2020) for studies involving humans.

**Informed Consent Statement:** Informed consent was obtained from all subjects involved in the study.

**Data Availability Statement:** The data presented in this study are available on request from the corresponding author.

**Acknowledgments:** The authors would like to thank the participants for taking part in this study. Patrick Connolly (MED-EL) edited a version of the manuscript.

**Conflicts of Interest:** The authors declare no conflicts of interest.

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
