# Peer review of "Objective and Subjective Assessment of Music Perception and Musical Experiences in Young Cochlear Implant Users"

_audiolres, doi:10.3390/audiolres14010008_

Round 1

Reviewer 1 Report

Comments and Suggestions for Authors

The manuscript titled “ Objective and subjective assessment of music perception and  musical experiences in young cochlear implant users” that I have had  the pleasure of reviewing, is very interesting and certainly touches on an important aspect of cochlear implant research. The authors through compositional measures investigate the ability of cochlear implant children and adolescent to perceive differences between instrumental musical pieces and between musical instruments compared to normal hearing pairs.

Although overall it is well written, I found some critical issues that I will specify in points below.

INTRODUCTION: I recommend expanding the introduction more by specifying the importance of multidisciplinary research in the area of music perception through cochlear implants . Specifically, I would suggest integrating the references in the literature with, for example, the following references.

The comprehensive and very interesting review  McDermott, H. J. (2004). Music perception with cochlear implants: a review. Trends in amplification8(2), 49-82.

Experimental studies with a multidisciplinary approach such as Cartocci, G., 

Inguscio, B. M. S., Mancini, P., Greco, A., Nicastri, M., Giallini, I., Leone, C. A., ... & Cartocci, G. (2022). ‘Musical effort’and ‘musical pleasantness’: a pilot study on the neurophysiological correlates of classical music listening in adults normal hearing and unilateral cochlear implant users. Hearing, Balance and Communication20(2), 79-88.

LINES 43-50: Provide more support for the research objective: why it is important for cochlear implant recipients to be able to differentiate vocal, instrumental music and instruments?

MATERIAL AND METHODS: LINE 58: Insert reference for criterion for inclusion of facility under 3 years old and for the age criteria inlusion (6-16)-

LINES 75-77: Include references in the literature on music selection

LINE 80: “Prior to the start of the study, each participant had to show us which songs they were familiar with”: How was such information recorded ? What method

LINE 83: The percentage of correctly 83 recognized songs was recorded for each participant.  ? How did you record that information ? Pen and paper ? Through software ?

LINE 85-88: Specify the experimental set up. How did the musical stimulation take place ? Through audio speakers ? Earphones ? What  was the direction of the sound ? Frontal, lateral ? Did it occur in the same way for both unilateral and bilateral cochlear implant recipients ?

LINE 103: Insert references on how the questionnaire was constructed

What were the durations of the audio tracks ?

Might it be very  helpful for understanding to insert a figure of the experimental protocol.

Table 3: not understandable

Insert the limitations of the study and possible applications of the results found.

Author Response

Thank you very much for taking the time to review this manuscript. Please find the detailed responses below and the corresponding revisions/corrections highlighted in the re-submitted files.

Reviewer 2 Report

Comments and Suggestions for Authors

Thank you for submitting this manuscript.  I recognize how difficult it can be to have assessment paradigms (subjective and objective) to assess similarities and differences between CI users and those with normal hearing (NH).  I would like to have seen a bit more discussion starting on line 100 with the introduction of the MuQPP questionnaire.  For example, why did the questionnaire use the questions that it did?  Also, would it be possible to have the questionnaire translated into English to be used in an appendix?

One line 124, you noted that the distribution was nor normal.  Perhaps a few lines would be in order here to talk about the distribution that you did find as well a perhaps why you didn't (or couldn't) transpose the data using some function such as logx or exp(x)? This would strengthen your argument to use non-parametric statistics such as the Mann Whitney U test.

In Table I, almost all of the children (13/14) and adolescents (15/16) were fit bilaterally.  Do you think that your results would have been any different if they had been fit bimodally.  There is some evidence that a bimodal fitting would be better (for speech and music perception). Comments on this?

Lines 162-167:  You found that CI recipients were worse than NH children to identify the clarinet, flute and violin.  This is odd in the sense that these are quite different instruments- a quarter wavelength reeded instrument (clarinet), a half wavelength non-reeded woodwind (flute), and a half wavelength stringed instrument (violin).  I would have expected that there was a difference between bass vs. non-bass instruments as found in some other studies that you had referenced).  In the lines 252-257 area of the manuscript I wonder about extending this discussion to try to account for these statistically significant results (as well as those differences that did not achieve statistical significance)?

Author Response

(The authors gave the same response as above.)

Round 2

Reviewer 1 Report

Comments and Suggestions for Authors

Dear Authors, 

my comments have been correctly interpreted and integrated into the manuscript. 

kind regards

Reviewer 2 Report

Comments and Suggestions for Authors

Thank you for your comments and responses.